# Putting It All Together: The Roles of Ribosomal Proteins in Nucleolar Stages of 60S Ribosomal Assembly in the Yeast *Saccharomyces cerevisiae*

**DOI:** 10.3390/biom14080975

**Published:** 2024-08-09

**Authors:** Taylor N. Ayers, John L. Woolford

**Affiliations:** Department of Biological Sciences, Carnegie Mellon University, Pittsburgh, PA 15213, USA

**Keywords:** ribosome assembly, protein–rRNA interactions, rRNA folding, ribosome assembly intermediates, ribosomal proteins, assembly factors, large ribosomal subunit, pre-rRNA processing, cryo-electron microscopy

## Abstract

Here we review the functions of ribosomal proteins (RPs) in the nucleolar stages of large ribosomal subunit assembly in the yeast *Saccharomyces cerevisiae*. We summarize the effects of depleting RPs on pre-rRNA processing and turnover, on the assembly of other RPs, and on the entry and exit of assembly factors (AFs). These results are interpreted in light of recent near-atomic-resolution cryo-EM structures of multiple assembly intermediates. Results are discussed with respect to each neighborhood of RPs and rRNA. We identify several key mechanisms related to RP behavior. Neighborhoods of RPs can assemble in one or more than one step. Entry of RPs can be triggered by molecular switches, in which an AF is replaced by an RP binding to the same site. To drive assembly forward, rRNA structure can be stabilized by RPs, including clamping rRNA structures or forming bridges between rRNA domains.

## 1. Introduction

Ribosomes are ribonucleoprotein nanomachines that translate the genetic code in mRNA and catalyze peptide bond formation. The small ribosomal subunit (SSU) houses the mRNA decoding center, while the large ribosomal subunit (LSU) contains the peptidyltransferase center (PTC). In yeast, the SSU is comprised of the 18S rRNA (1800 nucleotides (nts)) and 33 different ribosomal proteins (RPs). The LSU contains the 5S rRNA (121 nts), 5.8S rRNA (158 nts), 25S rRNA (3396 nts), and 46 different RPs.

The LSU 25S rRNA is organized into six phylogenetically conserved domains of secondary structure (domains I–VI, 5′ to 3′) that intertwine with each other (Appendix A) [1,2]. Each rRNA domain originates from a root helix created by base-pairing between the first and last nucleotide sequences of that domain, defining the domain “boundaries” (Appendix A). The 5.8S rRNA base-pairs with the 5′ end of rRNA domain I to form the proximal stem (Appendix A), while the 5S rRNA docks onto rRNA domain V to form the central protuberance, a bridge-like structure that links the two subunits during protein synthesis. RPs decorate the outer surface of rRNA and burrow into the rRNA core to help knit together the multiple rRNA domains (Appendix A).

Assembly of the ribosomal subunits begins in the nucleolus, with transcription of the 35S pre-rRNA by RNA polymerase I and the 5S rRNA by RNA polymerase III (Appendix A). The 35S primary rRNA transcript contains the sequences for the mature 18S, 5.8S, and 25S rRNAs, which are separated and flanked by internal and external transcribed spacers (Appendix A). During, and immediately after its synthesis, rRNA undergoes folding, modification, and removal of some of the spacer sequences, accompanied by binding of a subset of the RPs (Appendix A) [3,4,5,6]. The 5S rRNA is separately packaged into a 5S ribonucleoprotein complex (RNP) before joining nascent 60S subunits [7,8,9]. Small subunit precursor particles are released from the transcription complex upon endonucleolytic cleavage of rRNA at the A_2_ site, leading to rapid transit of pre-40S particles out of the nucleolus (Appendix A). In contrast, free pre-60S particles are only generated upon termination of rRNA transcription and then undergo multiple remodeling steps before moving into the nucleoplasm (Appendix A). Additional restructuring of precursors to both subunits occurs in the nucleoplasm, followed by final stages of maturation and quality control in the cytoplasm [10,11,12,13]. In rapidly growing yeast cells, 2000 ribosomes are produced each minute [14]. The accuracy and efficiency of this complex pathway of ribosome biogenesis are enabled at different steps by more than 200 AFs in yeast (Appendix A) [3,5,6]. Of these AFs, approximately 90 participate in LSU biogenesis. 

## 2. rRNA Folding Is a Critical Component of Ribosome Assembly

Arguably, the most challenging task of assembling ribosomes is the folding of their rRNA [15]. Because functional centers of the ribosome are largely comprised of rRNA, e.g., the peptidyltransferase center and the polypeptide exit tunnel in the LSU, proper formation of the three-dimensional structure of rRNA is especially critical for ribosome function [16]. 

RNA structure probing and near-atomic resolution cryo-electron microscopy (cryo-EM) structures of ribosome assembly intermediates have provided an initial glimpse into the rRNA folding pathway [5,6,17,18,19,20]. As rRNA is transcribed, it begins to fold into its secondary structure and bind to RPs and AFs (Appendix A). Yet the path to do so is not always straightforward. The complementary strands of some rRNA helices in mature ribosomes, including the root helices at the base of each domain, are far apart in the rRNA primary sequence (Appendix A). Therefore, alternative rRNA helices can form initially via base-pairing with other sequences 5′ of the mature partner before the second strand of that root helix is even transcribed. These initial helices then must be remodeled to promote productive pairing between downstream rRNA partners [21,22,23,24,25]. Once rRNA secondary structures are established, the resulting helices then reorient to form the complex tertiary structures necessary for rRNA function. 

One challenge to defining the rRNA folding pathway (for both cells and experimentalists!) is that a significant fraction of the rRNA is structurally heterogeneous during certain stages and thus cannot be resolved via cryo-EM. For example, newly synthesized rRNA is highly flexible in the earliest assembly intermediate, with less than one fourth of the LSU rRNA visible in the structure of the Noc1-Noc2 particle [26,27]. Likewise, most of rRNA domains III, IV, and V are flexible and do not stably compact onto pre-ribosomes until late nucleolar stages, after the installation of domains I, II, and VI (Appendix A) [18,20,28]. Thus, it remains unclear precisely how and when these initially flexible RNAs fold into secondary and tertiary structures and bind to RPs and AFs.

## 3. How to Determine Functions of Ribosomal Proteins

The primary function of RPs and many AFs is to enable the proper folding of rRNA. This facilitates the recruitment of other RPs and AFs to promote further remodeling of the pre-rRNP structure to drive assembly forward [6,29]. Before reviewing the roles of RPs in LSU assembly, we will first describe the experimental approaches used to assay RP functions. 

### 3.1. Investigating When Each RP Assembles into Pre-Ribosomes

Determining the order of RP assembly into pre-ribosomes provides useful hints for delineating the specific role of RPs in subunit maturation. Thirty-nine of the forty-six RPs in the large subunit associate with pre-ribosomes in the nucleus, while the remaining seven join nascent 60S subunits in the cytoplasm (Appendix A) [30]. To more precisely determine when each RP assembles, the consecutive pre-rRNA processing intermediates (Appendix A) and pre-ribosome assembly intermediates (Appendix A) with which each RP co-immunopurifies have been identified [31,32,33,34,35,36]. However, these approaches provide only rough estimates of RP entry times since there are only a handful of different pre-rRNA processing intermediates or purified pre-ribosome assembly intermediates that one can distinguish (Appendix A). Pre-ribosomes are captured using epitope-tagged AFs that copurify with multiple different assembly intermediates that span early, middle, and/or late intervals. Thus, affinity-purified pre-ribosomes usually comprise a heterogeneous population. In this review, we indicate the point of entry for each RP based upon when it is first resolved in pre-ribosome intermediates via cryo-EM. Note that biochemical assays and cryo-EM structures of pre-ribosomes do not always agree. Failure to visualize a protein in a particle may result from the structural heterogeneity and/or flexibility of that protein.

### 3.2. Depleting Ribosomal Proteins

Of the 46 RPs in the yeast LSU, 35 are essential for cell growth, while 11 are not essential under laboratory conditions [30,37]. However, deletion of the “nonessential” RP genes usually results in slow growth at 30 °C or an inability to grow at lower temperatures (cold sensitivity) [37,38]. Conditional expression has been utilized to deplete RPs whose absence causes a lethal or severe growth defect at 30 °C. To deplete an RP, a glucose-repressible *GAL* promoter is fused to the RP gene of interest. Cell cultures shifted from galactose- to glucose-containing medium halt transcription of the RP mRNA [39]. After waiting for the pre-existing RP mRNA to decay, and for the pool of unassembled RPs to be depleted by turnover or by assembly into pre-ribosomes, the depletion phenotype can be assessed. Phenotypes of cold-sensitive knockouts of nonessential RP genes can be assayed by shifting cultures to low temperatures.

To date, strains conditional for the expression of 42 of the 46 yeast LSU RPs have been constructed. Most of these mutants exhibit a shortage of 60S subunits relative to 40S subunits and are blocked at a particular stage of pre-rRNA processing, although some arrest assembly occurs only after all steps of rRNA processing have occurred [30]. The effects of the depletions on subunit maturation have been examined in more detail for 28 of the RPs. To do so, semi-quantitative mass spectrometry was used to assay changes in protein constituents of affinity-purified pre-ribosomes [40].

Interpreting RP depletion phenotypes is not necessarily straightforward. Depletion of an RP using conditional promoter systems can take several hours relative to the 15–20 min required for subunit assembly [40,41,42]. Thus, one must be aware of indirect effects. Defects in early steps of assembly can result when a late step is blocked by RP depletion due to improper recycling of early-acting AFs from the abortive late intermediates [42,43]. RPs could also impact assembly even before they form stable complexes with their rRNA ligands by first chaperoning the binding of other RPs [44]. Conversely, an assembly defect might only be observed downstream of the stable installation of an RP with an rRNA domain if that entire domain is initially flexible and then stably incorporated into pre-ribosomes at a later step [7]. Finally, an RP may be important for multiple stages of subunit maturation, yet its depletion may only reveal the earliest defect. For example, an RP may stably contact multiple neighborhoods of rRNA, with the potential to affect the assembly of more than one rRNA domain at different stages of biogenesis [45]. 

## 4. Roles of RPs in LSU Assembly

Here, we review the roles of RPs in the nucleolar stages of LSU biogenesis revealed by biochemical and genetic approaches in yeast and speculate how recently obtained cryo-EM structures of LSU assembly intermediates might help us better imagine the mechanistic principles underlying ribosome assembly. We discuss RPs whose association with pre-ribosomes has been detected in the nucleolus, including three RPs that are not visualized in particles by cryo-EM until the nucleoplasm (L5, L11, and L39). We assign each RP to one RNP domain in the LSU based on the rRNA with which that RP establishes the most contacts (Figure 1). However, we note that in many cases, RPs have significant contacts with more than one rRNA domain. The construction of each “neighborhood” in the LSU is discussed based on the order of their assembly: rRNA domain I, rRNA domain II, the proximal stem, rRNA domain VI, rRNA domain III, rRNA domain IV, rRNA domain V, the polypeptide exit tunnel, and the central protuberance. This order is not strictly 5′ to 3′ with respect to the rRNA sequence; domains at the two ends of the pre-rRNA (domains I, II, and VI) destined for the LSU assemble before those in the middle portion (domains III, IV, and V) (Figure 1, Appendix A) [18,46]. 

### 4.1. rRNA Domain I: L8, L13, L15, L36

In mature ribosomes, RPs L8, L13, L15, and L36 primarily contact rRNA domain I (nucleotides 1–651) (Appendix A and Figure 1B), but they also form bridges with neighboring domains II, V, and 5.8S rRNA (Appendix A). Most likely, these four RPs assemble co-transcriptionally, as they are first visible in the cryo-EM structure of the co-transcriptional Noc1-Noc2 particle (Figure 2A) [26]. L8, L13, and L15 associate with rRNA domain I when only this fragment of rRNA is expressed, while L36 only assembles once a construct containing both domains I and II is expressed [48]. The so-called “A_3_” AFs Brx1, Cic1, Has1, Ebp2, Erb1, Nop7, Nop15, and Rlp7 form a shell surrounding the RPs bound to domain I rRNA, and also assemble early, in the Noc1-Noc2 particle (Figure 2A). 

The effects of depleting L8, L15, and L36 have been investigated, but not yet for L13 [32,40,46,49]. Primer extension assays have revealed the accumulation of modest amounts of pre-rRNAs with 5′ ends at the A_2_, A_3_, and B_L_ sites. Northern blotting showed increased levels of 35S and 23S pre-rRNAs, while amounts of other downstream processing intermediates and mature rRNAs were significantly decreased. Consistent with these steady-state assays, pulse-chase experiments showed that the 27S pre-rRNAs are rapidly turned over. Thus, blocking assembly of domain I results in disassembly and degradation of early assembly intermediates. 

Analysis of pre-ribosome protein constituents following domain I RP depletions suggested that assembly of the four domain I RPs is largely, but not completely, hierarchical (Figure 2B). L13, L15, and L36 are the RPs most strongly diminished in the absence of L8, while L13 is affected by the depletion of L15 or L36, but L8 is much less so. However, this pattern of linear dependence does not appear to be consistent with the structure of domain I in the mature LSU. L15 is located on the interior of domain I embedded within domain I rRNA, while L8 is located on the outer surface bound to L15 (Figure 2C). Both L8 and L15 contact distant rRNA helices 10 and 15, appearing to pinch them together. In this configuration, we imagine that L8 would depend upon L15 to assemble, as well as vice versa. For example, we suspect that because L15 associates with numerous domain I rRNA regions, L15 could help the rRNA fold co-transcriptionally. This chaperoning could facilitate L8 binding, which requires the coming together of distal secondary structure elements, ultimately stabilizing the proximity of helices 10 and 15 in three dimensions. Thus, the seeming contradiction between molecular assays and existing structural snapshots suggests that the pathway of assembly may be more complex than initially evident from depletion experiments. This example signifies the need for more sophisticated real time assays of subunit maturation (see below in Section 6). 

Depletion of L8, L15, or L36 affects steps upstream and downstream of their assembly. Amounts of the very early scaffolding AFs Nop4, Rrp5, Noc1, and Noc2 increase in the population of purified pre-ribosomes, suggesting that early stages of assembly cannot proceed efficiently when domain I cannot properly form. Alternatively, a pool of pre-existing earlier intermediates containing these scaffolding factors might remain in the depletion strains and could be more stable than later particles in which assembly of domain I is aborted. Meanwhile, there are decreased amounts of the A_3_ AFs surrounding the RPs bound to domain I (Figure 2A). Interestingly, the reverse is not true. Depletion of A_3_ factors does not affect the assembly of domain I RPs, consistent with these AFs coating the exterior surface of domain I RNP in pre-ribosomes [50,51]. The absence of L8, L15, and L36 also impairs the assembly of other RPs. Amounts of the proximal stem and domain III RPs L17, L26, L19, and L31 that join in downstream steps of subunit maturation strongly decrease, yet the effects on assembly of other RPs with rRNA domains II or III are much weaker. 

Strikingly, many RPs contain N- or C-terminal domains or internal loops that extend from their globular core. These extensions thread through neighboring rRNA domains across the surface of the ribosome or penetrate into the ribosome core [52]. Domain I RP L8 is one example of such a protein that contacts multiple rRNP domains. While the globular domain of L8 binds to rRNA domain I, the N-terminal extension bridges rRNA domain I with domain V and the proximal stem. Greater than half of the L8 contacts with rRNA occur via its N-terminal extension (Figure 2D).

Deletion of the L8 extension revealed that this portion of L8 participates in late stages of assembly that involve domain V [45]. A mutant RP L8 lacking the 70 N-terminal amino acids can assemble into pre-ribosomes, but it causes a dominant lethal growth defect. While depletion of L8 aborts early steps of pre-rRNA processing and subunit assembly, the *rpl8 Δ1-70* mutation blocks later steps: processing of the 7S pre-rRNA, a precursor of 5.8S rRNA, and assembly of AFs required for remodeling of domain V and the 5S RNP. Thus, binding of the globular domain of L8 with domain I rRNA is important for the construction of domain I in the early steps of LSU biogenesis, while insertion of its N-terminal extension into domain V plays a role in later steps of subunit maturation that are coupled with the compaction of domain V and 7S pre-rRNA processing.

### 4.2. rRNA Domain II: L4, L6, L7, L14, L16, L18, L20, L32, L33

Nine RPs have extensive contacts with rRNA domain II (nucleotides 652–1455) in mature LSUs (Appendix A and Figure 1C). Among these proteins, L4, L7, L18, and L32 form one “node” that bridges domain II with rRNA domain I, while L6, L14, L16, L20, and L33 form a separate node that contacts both rRNA domains II and VI (Figure 3A and Appendix A).

Domain II (as well as domain I) adopts its mature conformation very early; both domains are organized in the Noc1-Noc2 particle in a fashion that is very similar to that in mature LSUs [26,53]. During the transition from the Noc1-Noc2 particle to the State 2 intermediate, a molecular switch draws together domains I and II (Figure 3B). Rrp1 and Mak16 bind primarily to domain II in the Noc1-Noc2 particle, but Nsa1 and Rpf1 are only loosely tethered to this complex. Upon release of Noc1 and Noc2, Nsa1 and Rpf1 become stably accommodated into State 2 particles, working with Rrp1 and Mak16 to form a bridge connecting domains I and II. 

Anchoring of domain VI onto domain II, mediated by RP bridges, may function as a “lid” to protect the 5′ end of 5.8S rRNA from degradation by nucleases [34]. Node II RPs L14 and L16 bound to domain II RNA may function in this compaction of domain VI by acting as clamps to stabilize the docking of domains II and VI. The globular portions of L14 and L16 are adjacent to one another, as are contact domains II and VI. The C-terminal extensions of these RPs contact one another in the Noc1-Noc2 particle, similar to the mature structure, while the C-terminal domain of L16 also cradles domain VI rRNA in mature subunits (Figure 3C). Truncating the C-terminal portion of either of these proteins (*rpL14-Δ109*, *rpl16-Δ171*) results in lethality [34,54,55]. The truncated L16 Δ171 protein assembles into pre-ribosomes, but pre-rRNA processing is affected similarly to a depletion of L16. Thus, analogous to the behavior of L8 in domain I, the globular domain of L16 is sufficient for assembly of this RP, but the C-terminal extension plays an important role in LSU maturation.

The effects of depleting each of the domain II RPs have been examined, except for L6 [32,34,40,54,55]. In each case, as observed for depletion of RPs present in domain I, amounts of 27SA_2_ and 27SA_3_ pre-rRNAs increase slightly, then are rapidly degraded. When domain II RPs are depleted, the association of RPs with domain I is less affected than for domain II RPs, and vice versa. Thus, the assembly of rRNA domains I and II is somewhat independent of each other, consistent with their initial physical separation from each other in Noc1-Noc2 particles (Figure 3B). However, in each depletion, abortive pre-ribosomes rapidly disassemble, and the pre-rRNA is degraded.

When domain II RPs are depleted, the amounts of RPs and AFs in pre-ribosomes that assemble downstream of domain II are diminished. This likely results from the turnover of particles containing 27SA pre-rRNAs, preventing the construction of later intermediates containing 27SB pre-rRNA. However, as observed for depletion of domain I RPs, a subset of RPs in the proximal stem (L17 and L26) and in domain III (L19 and L31) are consistently affected more than others (see discussion in Section 4.5 below).

Assembly of AF Nsa1 is also decreased upon depletion of domain II RPs. This probably results from preventing the transition of Nsa1 from a flexible state, loosely associated with pre-ribosomes, to being stably installed into State 2 particles following the exit of AFs Noc1 and Noc2 (Figure 3B) [26].

Binding of the AF Rrp1 to pre-ribosomes is also strongly affected by depletion of L4, L18, or L32. L18 and the globular body of L4 appear to form the binding site for Rrp1 in the Noc1-Noc2 particle (Figure 3D). Thus, Rrp1 relies on the accurate incorporation of L4 and L18 into pre-ribosomes. However, Rrp1 does not appear to make direct contact with L32. We speculate that the absence of L32 disrupts the local structure of this neighborhood, impairing the binding site for Rrp1 [26].

### 4.3. Proximal Stem: L17, L26, L35, L37

In mature 60S subunits, the proximal stem is formed by base-pairing portions of 5.8S rRNA with nucleotides in domain I of 25S rRNA (Appendix A, Figure 1D and Figure 4A,B). RPs L17, L26, L35, and L37 bind to rRNA in the proximal stem but also contact multiple other domains of rRNA (Figure 1D, Figure 4A and Appendix A). L17 binds to helix 2, formed by base-pairing the very 5′ end of mature 5.8S rRNA with 25S rRNA, while L26 is positioned downstream bound to helix 4. The globular domain of L35 contacts folded 5.8S rRNA, while its C-terminal extension snakes into domain I. Similarly, L37 binds 5.8S rRNA but also bridges domains I and III.

The proximal stem begins to form during the early stages of LSU assembly. Nucleotides in 5.8S rRNA base-pair with those in 25S rRNA co-transcriptionally, visualized in Noc1-Noc2 particles [26]. Formation of these base-pairs creates the pre-ribosome “foot structure” containing the ITS2 spacer RNA sequences that lie between 5.8S rRNA and domain I in the 27S pre-rRNA (Figure 4B). Subsequent nucleolytic removal of ITS2, at much later steps of LSU assembly in the nucleoplasm, creates the 3′ end of mature 5.8S rRNA and the 5′ end of mature 25S rRNA, whose proximal sequences base-pair with each other to form helix 10 (Figure 4B). 

While the proximal stem rRNA structure is formed early, all four proximal stem RPs are visualized in pre-ribosomes only after assembly of rRNA domains I and II, upon conversion of the Noc1-Noc2 particle to the State A intermediate (Figure 4C). Their incorporation during this transition is enabled by a molecular switch (Figure 4D) [26]. In the Noc1-Noc2 particle, Noc1 encircles helix 2. Upon release of Noc1, L17 binds to helix 2. Likewise, the exit of Noc2 bound to helix 25 enables the incorporation of L26 as well as AFs Nsa1 and Rpf1. 

Although ribosomal protein L26 is evolutionarily conserved, it is one of 11 yeast RPs in the LSU that is not essential for cell growth [37]. Deletion of both *RPL26* genes has only mild effects on pre-rRNA processing, the composition of pre-ribosomes, or the transit of pre-ribosomes from the nucleus to the cytoplasm [56]. In contrast, L17, L35, and L37 are essential RPs. Their depletion strongly blocks processing of 27SB pre-rRNA, leading to turnover of the rRNAs, although more slowly than occurs for 27SA_2_ and 27SA_3_ pre-rRNA when domain I or II RPs are depleted [32,46,57,58]. 

The proximal stem RPs do not appear to depend on each other to stably associate with pre-ribosomes. Depletion of any one of them does not strongly affect the assembly of the other three. In contrast, downstream assembly of other RPs and AFs is diminished, most likely as a result of the turnover of the abortive earlier intermediates. Again, as with domain I and II RP depletions, the assembly of L19 and L31 is consistently affected significantly more than that of the other six RPs that contact domain III (see discussion below in Section 4.5). 

### 4.4. rRNA Domain VI: L3, L9, L23

rRNA domain VI (nucleotides 2995–3396), (Appendix A, Figure 1E and Figure 5A) is stably accommodated into pre-ribosomes in a stepwise fashion during the early stages of large subunit maturation [18,20]. Approximately 22 percent (88/401 nucleotides) of the rRNA in domain VI becomes visible in State A particles, while the remainder of domain VI can first be seen by cryo-EM in State 2 particles (Appendix A and Figure 5A). RPs L3 and L23 primarily contact domain VI rRNA and are first visible in State 2 particles (Figure 1E and Figure 5A). L9 interacts with rRNA in both domains IV and V, as well as domain VI, and is first visible in State C assembly intermediates (Appendix A, Figure 1E and Figure 5A). 

Pulse-chase and steady-state assays of L3-depleted cells reveal only a minor effect on amounts of 27SA_2_ pre-rRNA compared to wild-type cells. In contrast, levels of 27SB pre-rRNA are strongly reduced [32,46,59]. Consistent with this early block in subunit maturation, the amounts of RPs and AFs recovered in affinity-purified pre-ribosomes are decreased in all but the earliest assembly intermediates. These strong effects upon L3 depletion might result from the fact that L3 functions early in assembly, together with the Npa1 complex of AFs, to bind to and consolidate rRNA from multiple domains [60]. In mature subunits, L3 is proximal to the clustered root helices of all six rRNA domains, as well as the 5′ end of 5.8S rRNA and the 3′ end of 25S rRNA (Figure 5B).

Depletion of either L9 or L23 blocks a later step in subunit maturation than when L3 is depleted and results in slower turnover [32,46]. However, L23 and L9 are important for different steps of assembly. Upon depletion of L23, the formation of State 2 particles is blocked, resulting in decreased amounts of AFs Tif6, Rlp24, and Nog1 that first enter these State 2 particles (Appendix A) and in decreased amounts of proteins that associate with downstream intermediates. Tif6 binds directly to L23, while Rlp24 and Nog1 are proximal to L23 but do not contact it directly. Without proper incorporation of L23, it is possible that local RNA structure is perturbed, impacting binding of Rlp24 and Nog1 (Figure 5C). Interestingly, during the remodeling of helices 90–92 in the PTC by the RNA helicase Dbp10, L23 stabilizes both the initial alternate helix 92 as well as the subsequent mature helix 92 (Figure 5D) [23]. In both structures, this function may be enabled by the interaction of L23 with nearby domain VI rRNA. 

Depletion of L9 blocks the stable construction of State C particles. This is evident by the decreased amounts of AFs Nug1, Nsa2, and Dbp10 that enter pre-ribosomes during the transition of State 2 to State C (Figure 5E). Although the head domain of Nog1 lies near L9, assembly of this factor is not affected by L9 depletion, likely because the middle domain of Nog1 is anchored onto particles before entry of L9 [18].

### 4.5. rRNA Domain III: L19, L22, L25, L27, L30, L31, L34, L38

The nine RPs L19, L22, L25, L27, L30, L31, L34, L38, and L39 primarily contact rRNA domain III in mature 60S subunits (nucleotides 1456–1877, Appendix A, Figure 1F and Figure 6A). Because L39 seems to have completed its assembly and function after all of the other domain III RPs, it will be discussed in a later section (see below Section 4.8). L22, L25, L27, L30, and L38 bind to the exterior surface of domain III, while L19, L31, and L34 are embedded within domain III rRNA. The long C-terminal alpha-helix of L19 extends outward from the ribosome to form an inter-subunit bridge with 40S subunits during translation [47,61]. L31 bridges the root helix of domain III with domain VI rRNA, while the N-terminal extension of L25 threads from domain III into domain I. Deleting 61 amino acids from the N-terminus of L25 does not impact its ability to recognize its binding site in a fragment of 25S rRNA in vitro, suggesting that the C-terminal globular domain could be responsible for initial assembly onto the pre-ribosome [62].

Domain III rRNA is remodeled from an initially flexible state to stably compact onto the body of State D pre-ribosomes, where it is first visible (Appendix A) [18]. It is not yet completely clear precisely when RPs assemble with domain III rRNA, because this rRNA and any proteins bound to it are not visible by cryo-EM until State D particles (but see below).

Interestingly, of the 11 nonessential RPs in the LSU, L22, L30, L38, and L39 are bound to domain III rRNA. The *rpl22* knockout strain grows slowly at 30 °C and is unable to grow at low temperatures [37]. Depletion of the essential RPs L19, L25, L27, L31, or L34 has a strong pre-rRNA processing defect with the accumulation of 27SB pre-rRNA followed by its slow turnover [32,33,46,61,63]. 

Effects on pre-ribosome maturation have been examined upon depletion of L19 and L25, but not yet for other domain III RPs [33,63]. When either of these RPs is depleted, most of the proteins that assemble after the compaction of domain III fail to associate with pre-ribosomes. However, incorporation of many of the RPs that bind to domain III rRNA is not significantly affected, except for L19, L31, and L39. Similarly, assembly of L19, L31, and L39, but not that of the other domain III RPs, is also strongly diminished when maturation of domains I, II, VI, or the proximal stem is blocked during earlier stages of subunit maturation, before domain III is stably compacted and visible [34,46]. Taken together, these results suggest that L22, L25, L27, L30, L34, and L38 might associate with flexible domain III rRNA before it undergoes remodeling to compact into/onto the pre-ribosome. Their assembly may be independent of that of domains I, II, and VI during the early stages of LSU assembly and also might not depend on the completion of domain III construction at later stages of subunit maturation. Because assembly of L19 and L31 is coincident with compaction of domain III, and both proteins are juxtaposed with the root helices of domain III (helices 47, 48, and 60) (Figure 6A), it is tempting to speculate that compaction of domain III is driven by remodeling of its root helices coupled with insertion of L19 and L31. Alternatively, the formation of the mature root helix for domain III may simply create binding sites for L19 and L31. 

Interestingly, amounts of RPs bound to the proximal stem decrease when domain III RPs L19 and L25 are depleted, even though this neighborhood assembles upstream of domain III compaction (Appendix A). This apparent anomaly might be explained by the fact that the proximal stem RPs L17, L26, L35, and L37, together with domain III RPs L19, L25, and L31, are components of the exit platform of the polypeptide exit tunnel (Figure 6B). We speculate that failure to complete the tunnel exit platform by incorporation of L19, L25, and L31, and/or failure to compact domain III rRNP to complete the wall of the tunnel, might destabilize the platform and cause the RPs that had previously joined the platform to disassemble. 

### 4.6. rRNA Domain IV: L2, L43

Ribosomal proteins L2 and L43 bind to each other, sandwiched between rRNA domains II, III, IV, and V in mature ribosomes (Figure 1G and Figure 7A). Consistent with the extensive contact between L2 and L43, their assembly is mutually interdependent. These two RPs are first visible by cryo-EM in pre-ribosome State NE1, coincident with the transition from State E2 to NE1 (Figure 7B, left and center panels) [21,28]. The previously flexible portion of rRNA domain IV can now be visualized compacted onto rRNA domains II and III, stabilized by contacts of L2 and L43 with all three of these domains (Figure 7B, center panel). In the subsequent State NE2, the L1 stalk of rRNA domain V now becomes visible, compacted onto the body of the particle to contact L2 and L43 (Figure 7B, right panel). 

This cascade of rRNA compaction is triggered by the release of ten AFs from the E2 particle by the Rea1 AAA-ATPase (Figure 7C) [21,64]. This remodeling includes yet another molecular switch. When Noc3 and Rrp17 are released from the E2 particle, L2 and L43 can then bind to the same sites in the NE1 intermediate. The subsequent rotation of an Spb1 alpha-helix located above L43 appears to “lock down” L43 onto the particle, driving assembly forward by preventing re-entry of Rrp17 (Figure 7C). 

Both L2 and L43 co-immunoprecipitate significant amounts of 7S pre-rRNA and small amounts of 27SB pre-rRNA, suggesting that they assemble just before cleavage and removal of the ITS2 spacer RNA in 27SB pre-rRNA to produce 7S pre-rRNA [33]. Consistent with this timing of their entry into pre-ribosomes, depletion of L2 or L43 blocks this cleavage event, resulting in the accumulation of pre-ribosomes containing the three AFs that bind to ITS2 (Nop15, Rlp7, and Cic1) [32,65]. 

Importantly, L2 and L43 are the last RPs to associate with pre-60S particles before they transit from the nucleolus into the nucleoplasm. Assembly of these RPs was hypothesized to facilitate this exit of pre-ribosomes from the nucleolus (Figure 7C) [66]. Installation of L2 and L43 is coincident with the release of AFs Noc3, Ebp2, Erb1, and Nop16 that contain intrinsically disordered regions (IDRs), and with the compaction of flexible rRNA domain IV. The IDRs in these AFs and this flexible rRNA are hypothesized to form an interaction network between pre-ribosomes to create the nucleolar condensate [66]. Thus, these remodeling events, including the entry of L2 and L43, could decrease the potential of pre-ribosomes to interact with each other and therefore favor the release of pre-ribosomes from the nucleolar condensate. Indeed, depletion of L2 or L43 prevents the nucleolar exit of pre-ribosomes, whereas blocking the next step, maturation of NE2 particles to Nog2 particles, does not [65].

### 4.7. rRNA Domain V: L1, L21, L28

Both L21 and L28 primarily contact rRNA domains II and V on the top surface of the mature large subunit, directly underneath the central protuberance (Figure 1H and Figure 8A). L21 spans the solvent-exposed and inter-subunit surfaces, while L28 binds to the solvent-exposed surface. A helical portion of L21 is initially resolved in State D. Then, in State E1, the C-terminal domain of L21 is visible contacting domain II RPs L7 and L20 (Figure 8B) [21]. The C-terminal globular portion of L28 is first seen in State NE1, whereas the N-terminal domains of both L21 and L28 are not visible until the cytoplasmic stages of maturation (Figure 8C) [19]. The trigger for entry of L28 into pre-ribosomes is another example of a molecular switch, where the AF Brx1 is removed by the AAA-ATPase Rea1 to free up the binding site for the C-terminal portion of L28. The C-terminal domain of RP L13 then clamps down on L28 in state NE1 to stabilize its binding (Figure 8C).

Depletion of L21 and L28 appears to primarily affect steps downstream in the nucleoplasm. Small amounts of 27SB pre-rRNA and much larger amounts of 7S pre-rRNA accumulate [32,33,46]. In addition, depletion of L21 results in modest accumulation of 25.5S and 6S pre-rRNAs. Levels of AFs released from pre-ribosomes in the nucleoplasm moderately increase in L21-depleted cells. The release of AFs Rsa4, Nog2, Rpf2, and Rrs1 that are proximal to L21 appears to be slightly delayed, suggesting a partial effect on the remodeling of domain V, including the rotation of the 5S RNP that sits above L21 (Figure 10). Upon depletion of L28, the binding of nuclear export factors Bud20 and Nmd3 to pre-ribosomes is slightly diminished, consistent with their entry after L28. L10, L24, L29, and L40 are the only RPs markedly affected by the absence of L21 and L28 (Figure 8D) [33,46]. Of these, L10, L29, and L40 lie proximal to L21 and L28. L10 and L24 assemble in the cytoplasm, and therefore their assembly may be precluded due to a block in nuclear export. 

Ribosomal protein L1 binds the loop formed by helix 77 in rRNA domain V to form the L1 stalk, located adjacent to the E site in mature 60S subunits. There, L1 participates in translation elongation by facilitating the release of E site tRNAs and translation factors [67]. L1 is first visualized in state E1/E2 particles prior to the rotation of the L1 stalk (Figure 8E) [21]. In the NE2 particle, the L1 stalk swivels to its mature position, yet L1 is not visible in this conformation (Figure 8E).

Depletion of L1 results in a modest increase in 7S pre-rRNA [32]. Consistent with this late phenotype, L1 co-immunopurifies with more 7S pre-rRNA than 27S pre-rRNA. 

A portion of the essential ribosome export factor Nmd3 binds to L1 and holds the L1 stalk in a closed conformation in late nuclear assembly intermediates [19,68]. Restraining this hydrophilic stalk onto the pre-ribosome body might be important for particles to efficiently travel through the constrained, hydrophobic environment of the nuclear pore complex. Nmd3 as well as the export factors Bud20 and Arx1 are efficiently recruited to pre-ribosomes lacking L1 [69], but the export factor Mex67/Mtr2 does not assemble. Exactly how L1 affects the assembly of this factor remains unclear. Nevertheless, the absence of L1 or truncation of the rRNA that binds to L1 results in inefficient nuclear export of pre-60S particles. Some fraction of mutant particles enter the cytoplasm and associate with polysomes. Yet, their enrichment with relatively smaller polyribosomes suggests a defect in translation elongation, consistent with the role of L1 in this process. The effect of depleting L1 on the assembly of other RPS or AFs has not been reported.

### 4.8. The Polypeptide Exit Tunnel: L39

Ribosomal protein L39 is one of three RPs situated in the nascent polypeptide exit tunnel (NPET) of the 60S ribosomal subunit (Figure 1F and Figure 9A). L39 is embedded within helices 49–51 of domain III rRNA in the wall of the tunnel, near the tunnel exit site (Figure 9A). In eubacteria, the C-terminal extension of RP uL23 (a homologue of eukaryotic RP L25) is present adjacent to the NPET, but it is replaced by L39 in eukaryotes. Internal loops of two other RPs, L4 and L17, extend into the tunnel proximal to the PTC to form the so-called “constriction site”. L39 reduces the diameter of the NPET right at the exit of the tunnel, which could affect protein synthesis (Figure 9B). Indeed, L39 is important for the folding of nascent polypeptides to minimize their aggregation [38].

L39 is first visualized fully accommodated into the NPET of pre-ribosomes in Nog2 particles at the late nuclear stages of subunit maturation (Figure 9C) [70]. However, L39 appears to initially begin loading onto pre-ribosomes at State C or earlier. L39 copurifies with pre-ribosomes containing the AF Ssf1 that exits before State D particles, and partial densities of L39 are present in cryo-EM structures of E and NE1 particles [38]. The inability to visualize all of L39 in pre-ribosomes at these early stages of subunit assembly may result from its incomplete accommodation into the particle, leading to a flexible conformation. Thus, L39 may begin to assemble during or soon after the compaction of domain III, but it is not completely accommodated into pre-ribosomes until later. 

While L39 is not essential under normal growth conditions, *rpl39Δ* mutants grow very slowly in the cold and are defective in 60S subunit assembly [38]. In the absence of L39, small amounts of 27SB pre-rRNA and much larger amounts of 7S pre-rRNA accumulate, then both undergo slow turnover. This effect on processing of both 27SB and 7S pre-rRNAs is observed in other mutants blocked during late nucleolar stages, e.g., upon depletion of L21, L28, Nog2, or Rsa4 [32,71,72]. 

Despite the fact that L39 binds to domain III rRNA, the absence of L39 does not appear to impact the compaction of domain III, likely because the protein does not stably associate with pre-ribosomes until after this compaction has occurred (Figure 9C) [70]. Consistent with this idea, the stable installation of L39 is necessary for later nucleoplasmic steps of subunit assembly. In *rpl39Δ* knockout strains, AFs that are required for rotation of the 5S RNP and for nuclear export of pre-60S particles assemble much less efficiently. Interestingly, this same phenotype is observed in *rpl4Δ63-87* mutants where the RP L4 tunnel constriction site is deleted. L39 fails to assemble in this *rpl4Δ63-87* mutant, but assembly of L4 is not perturbed in the *rpl39Δ* mutant [38,73]. Thus, this failure of L39 assembly may account for the *rpl4* mutant phenotype.

### 4.9. Central Protuberance: L5, L11

RPs L5 and L11 bind to 5S rRNA to form the central protuberance (CP). This conserved structural feature of LSUs includes helices 80 and 82–88 of rRNA domain V situated on top of mature large ribosomal subunits, just above the peptidyl-transferase center (Appendix A and Figure 1I). 

As stated earlier, 5S rRNA is transcribed separately from the 35S precursor of the mature 18S, 5.8S, and 25S rRNAs. The 5S rRNA is assembled together with RPs L5 and L11 onto pre-ribosomes at an early, nucleolar stage of subunit maturation in particles containing 27SA_2_ pre-rRNA [7]. First, the dedicated chaperone Syo1 binds L5 and L11 in the cytoplasm and facilitates their import into the nucleus [74]. L5 and L11 join 5S rRNA and AFs Rpf2 and Rrs1 to form the 5S rRNP, which incorporates together into pre-ribosomes [7,8]. These four proteins and 5S rRNA, as well as helices 80 and 82–88 of domain V, are present but not visible by cryo-EM in the early assembly intermediates. Thus, the nascent CP, including portions of domain V, is initially flexible (Figure 10). Surprisingly, when the CP is first visible in Nog2 particles by cryo-EM, it is rotated almost 180 degrees backwards from its mature conformation, linked to domain V by Rpf2 and Rrs1. Release of Rpf2 and Rrs1 then destabilizes the pre-rotation state, helices 80 and 82–88 undergo remodeling, and additional AFs, including the AAA-ATPase Rea1, join the particle to stabilize the rotated state of the CP [75,76]. Subsequent ATP hydrolysis by Rea1 activates the release of Nog2 and other AFs to enable binding of the nuclear export adaptor Nmd3 and trigger the export of pre-60S subunits from the nucleus to the cytoplasm. 

Depletion of L5, L11, Rpf2, or Rrs1 prevents association of the 5S rRNP with pre-ribosomes [7]. Specifically, the C-terminal extensions of Rpf2 and Rrs1 are required for initial docking by tethering the RNP to early assembly intermediates [8]. Interestingly, preventing entry of the 5S RNP into pre-ribosomes does not block early steps of large subunit maturation, probably because the tethered, flexible 5S RNP does not physically impact other neighborhoods of the pre-60S particles [7]. Likewise, in mutants where the assembly of other rRNA domains is compromised during the early stages of subunit maturation, the 5S rRNP can still assemble into the mutant particles, before they are turned over [46]. Thus, the initial docking of the 5S RNP is independent of the assembly of other domains during the early stages of subunit maturation. The presence of the 5S RNP in pre-ribosomes is only important at later steps, when it is stably accommodated into Nog2 particles together with rRNA domain V (Figure 10). When L5, L11, Rpf2, or Rrs1 are depleted, 27SB and 7S pre-rRNAs accumulate, and the AF Nog2 fails to efficiently assemble. 

## 5. Discussion: Lessons Learned from RPs

### 5.1. An Ordered Pathway?

Unexpectedly, mapping the molecular phenotypes of RP depletions onto the crystal structure of yeast 60S subunits revealed that the 5′ and 3′ domains of rRNA (domains I, II, and VI) are assembled with RPs before middle domains III, IV, and V assemble into stable neighborhoods [46]. This was later confirmed by cryo-EM [18]. This pathway makes sense from the perspective of the 60S subunit structure: the solvent-exposed surface is built prior to the inter-subunit interface that contains the functional centers. However, the extent of alternate assembly pathways, as observed for bacterial ribosome subunit reconstitution in vitro [77], is not yet clear.

### 5.2. Individual Neighborhoods May Assemble in Discrete Stages

Portions of rRNA domains IV, V, or VI, including the proteins bound to them, are visualized by cryo-EM in a stepwise fashion (Figure 5A, Figure 7B and Figure 10). Similarly, domain III appears to assemble in several different steps. Even though domain III is largely invisible to cryo-EM in early stages, a subset of domain III RPs might bind to flexible domain III before the stable installation of the domain onto the body of the pre-ribosome. The subsequent compaction of domain III might be driven in part by the binding of the remaining RPs to the root helix of domain III (Figure 6). 

Functional centers also assemble in multiple steps. For example, the exit platform for the polypeptide exit tunnel is constructed in two separate stages. L17, L26, L35, and L37 bind to the proximal stem portion of the platform several steps before L19, L25, and L31 bind to the section comprised of a portion of domain III (Figure 6B and Appendix A). 

### 5.3. RPs Enable rRNA Folding 

Extended domains of RPs may function as bridges between rRNA domains. Globular domains of some RPs bind to one rRNA domain, while an N- or C-terminal extension of the same RP threads into an adjacent domain (e.g., L8, L16, and L25) (Figure 2D, Figure 3C and Figure 6A). Bridges between rRNA domains are also mediated by different portions of the globular domains of RPs (L2 and L43 with domains III, IV, and V, or L31 with domains III and VI) (Figure 6A and Figure 7A). RPs might also function as clamps to stabilize the binding of another RP, e.g., L13 with L28, and thus to drive assembly forward (Figure 8C).

RPs may not establish contact with all of their rRNA ligands in one step. In several cases, portions of an RP are not visible in the cryo-EM structure of assembly intermediates (e.g., L39). Yet, RPs can be detected by co-immunoprecipitation assays, although weakly associated with the rRNA. Such an RP may initially form “encounter complexes”, in which it searches multiple RNA conformations before stably binding its substrate [44]. Thus, depletion of such an RP can affect a stage of assembly before it is visible by cryo-EM. 

### 5.4. Molecular Switches

The entry of an RP may be dictated by the release of an AF that occupies the same binding site, e.g., Noc1 and L17, Noc2 and L26, or Rrp17 and Noc3 with L2 and L43 (Figure 4D and Figure 7C). 

## 6. Going Forward

Despite the recent near-atomic structures of assembly intermediates, we have only pre-ribosome snapshots, not movies. We lack a detailed understanding of the dynamics of ribosome assembly. Better approaches are needed to interrogate rRNA structure and rRNA folding. Exactly how does each RP establish all of its contacts with rRNA? How do the structures of the proteins and rRNAs change and affect each other during these encounters? What roles do RPs play in the compaction of flexible domains of rRNA? How are structural changes transmitted to proximal and distal rRNP neighborhoods in the pre-ribosome as assembly progresses? Single-molecule assays of RP binding and rRNA folding in real time, such as those carried out by the Woodson and Williamson labs for the assembly of individual domains of bacterial ribosomes, could shed light on these questions [44,78].

## Figures and Tables

**Figure 1 biomolecules-14-00975-f001:**
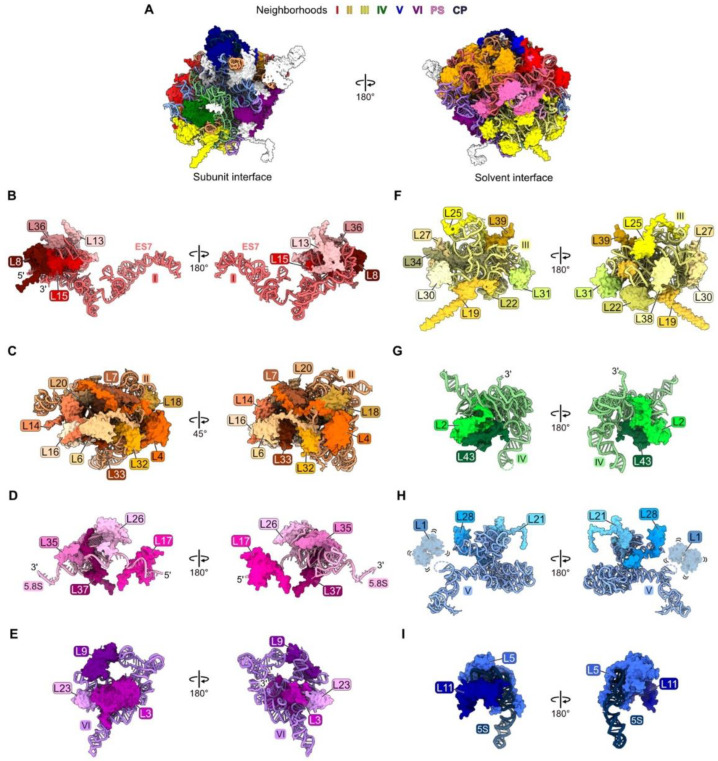
Each neighborhood of rRNA and RPs in the crystal structure of the yeast mature large ribosomal subunit [47], in the order with which they assemble: (**A**) The entire large subunit. Ribosomal proteins are color coded according to the neighborhood in which they are located. Ribosomal proteins colored in white assemble after nucleolar stages and thus are not discussed in this review; (**B**) domain I (red); (**C**) domain II (orange); (**D**) proximal stem, PS (pink); (**E**) domain VI (purple); (**F**) domain III (yellow); (**G**) domain IV (green); (**H**) domain V (blue); (**I**) central protuberance, CP (navy blue). Structures are shown in the subunit interface view on the left and the solvent-exposed surface on the right, except for domain II. In this case, orientations are shown to optimize the visualization of each RP. Note, though ribosomal protein L1 is present, it is not resolved in the mature LSU crystal structure (PDB: 4V88).

**Figure 2 biomolecules-14-00975-f002:**
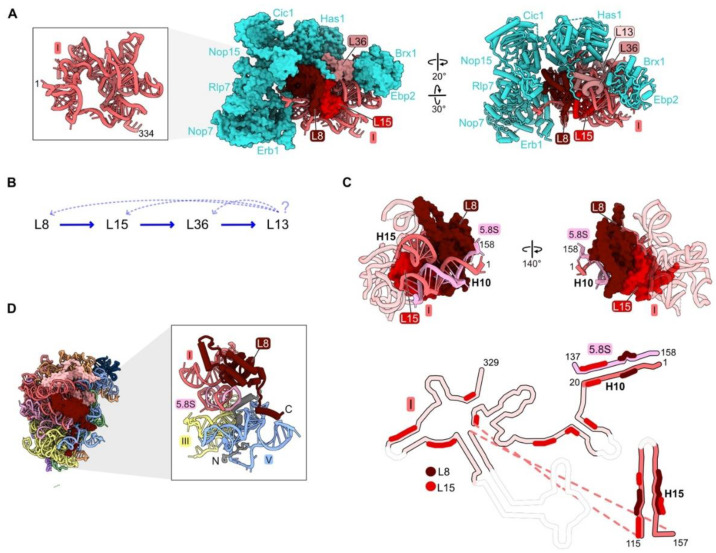
Domain I (RPs L8, L13, L15, L36): (**A**) Eight assembly factors (AFs) (cyan) bind to the exterior of the domain I RNP in the Noc1-Noc2 particle during early stages of assembly (PDB: 8E5T) [26]; (**B**) the assembly hierarchy implied by the effects of depleting RPs L8, L15, or L36; (**C**) binding of L8 and L15 to domain I rRNA and to each other in Noc1-Noc2 pre-ribosomes (PDB: 8E5T). Flexible rRNA is depicted as transparent in the secondary structure. Regions where L8 and L15 contact rRNA are indicated; (**D**) the globular domain of L8 (red) binds to rRNA domain I, while the 70-amino acid long N-terminal domain (grey) binds to rRNA domain V (PDB: 4V88).

**Figure 3 biomolecules-14-00975-f003:**
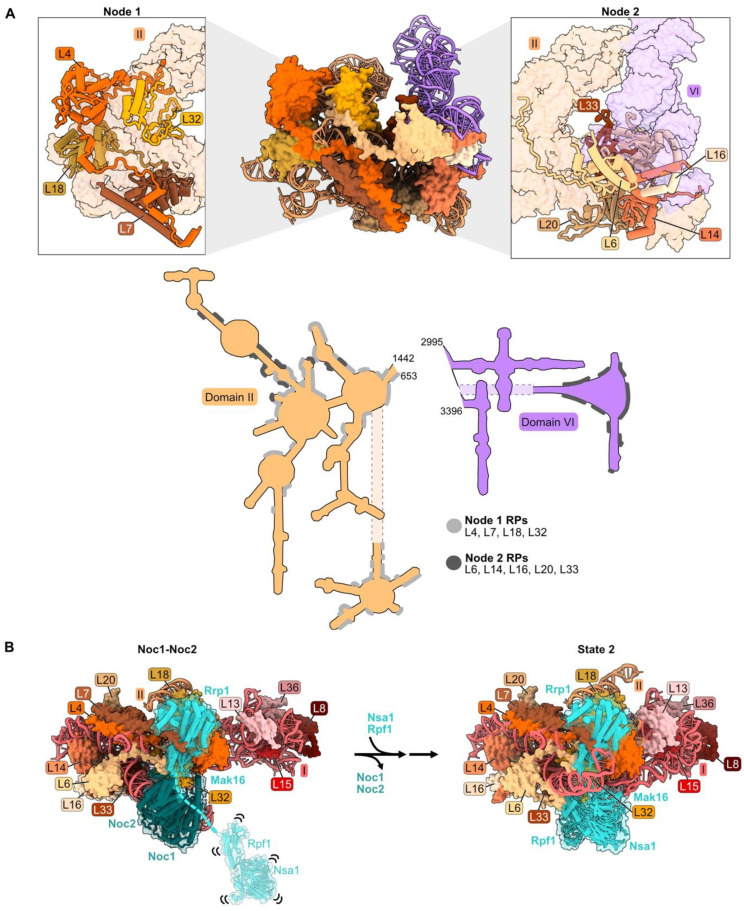
Domain II: (**A**) Structure of node 1 containing RPs L4, L7, L18, and L32 (left), and of node 2 containing RPs L6, L14, L16, L20, and L33 (right) (PDB: 4V88). Domain VI (purple) is shown docked onto domain II rRNA. The bottom panel depicts the contact points of each protein with the rRNA; (**B**) a molecular switch between the co-transcriptional Noc1-Noc2 particle and the State 2 intermediate (PDB: 8E5T, 6C0F) [20,26]. State A (lower resolution) precedes State 2 and is not shown; (**C**) docking of domain VI rRNA onto domain II rRNA during the transition from the co-transcriptional Noc1-Noc2 particle to the State 2 intermediate. Panels below show extensions of L14 and L16 binding to domain VI after compaction. Truncations of L14 and L16 C-termini (grey) result in lethality (PDB: 8E5T, 6C0F); (**D**) binding of AF Rrp1 to node 1 domain II RPs (PDB: 8E5T).

**Figure 4 biomolecules-14-00975-f004:**
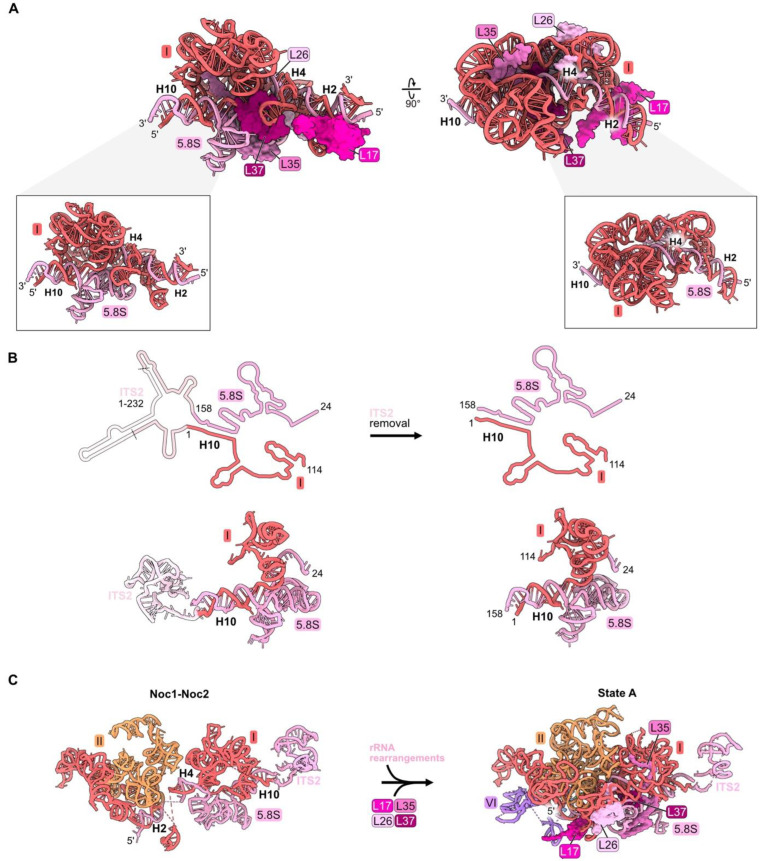
Proximal stem (RPs L17, L26, L35, and L37): (**A**) The structure of the mature proximal stem RNP (PDB: 4V88); (**B**) processing of the ITS2 spacer rRNA to create the mature proximal stem (PDB: 6C0F, 4V88); (**C**) Maturation of the proximal stem during the transition from the Noc1-Noc2 particle to the State A intermediate (PDB: 8E5T, 6EM3) [18,26]; (**D**) a molecular switch resulting in the replacement of AF Noc1 with RP L17 and AF Noc2 with L26, Nsa1, and Rpf1 during the transition from Noc1-Noc2 to State 2 (PDB: 8E5T, 6C0F).

**Figure 5 biomolecules-14-00975-f005:**
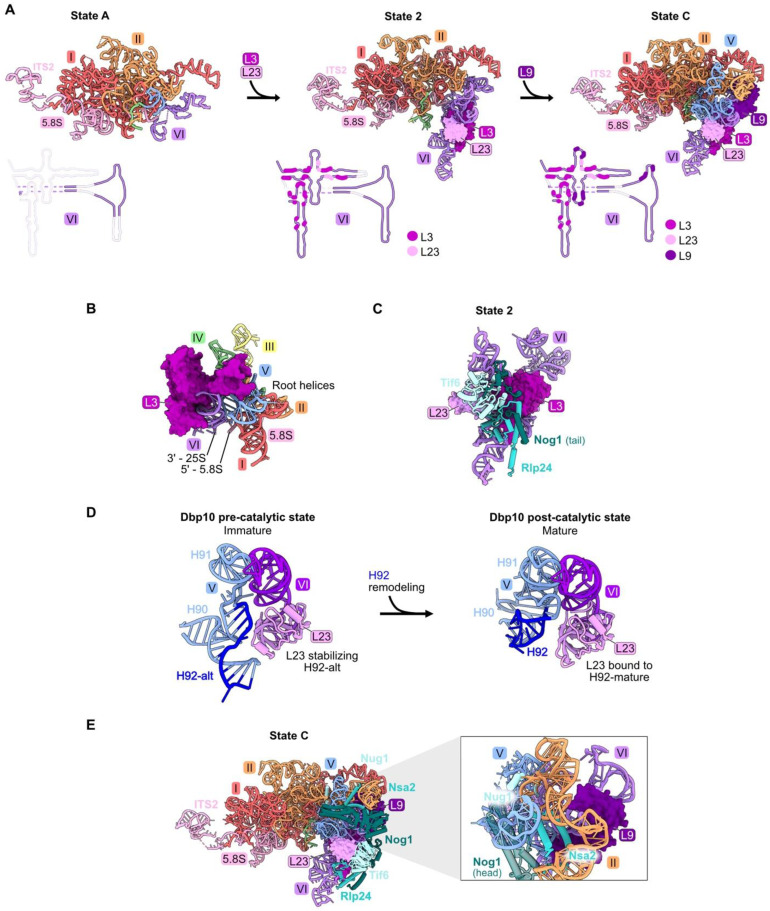
Domain VI (RPs L3, L9, and L23): (**A**) The sequential assembly of domain VI from State A to State C intermediates (PDB: 6EM3, 6C0F, 6EM1). Bottom panel: secondary structure diagrams indicating the successive stabilization of domain VI rRNA, including contacts with the domain VI RPs; (**B**) L3 contacts all six root helices, and is proximal to the 5′ end of 5.8S rRNA and the 3′ end of 25S rRNA in the mature LSU (PDB: 4V88); (**C**) the assembly of AFs Nog1, Rlp24, and Tif6 proximal to L23 is most affected by depletion of L23 (PDB: 6C0F); (**D**) the assembly of AFs Nug1, Nsa2, and Dbp10 adjacent to L9 is most affected by its depletion (PDB: 6EM1). Note that Dbp10 is invisible in these particles; (**E**) L23 stabilizes both conformations of H92 (H92-alt, then H92 in its mature form) (PDB: 8V83, 8V87) [23].

**Figure 6 biomolecules-14-00975-f006:**
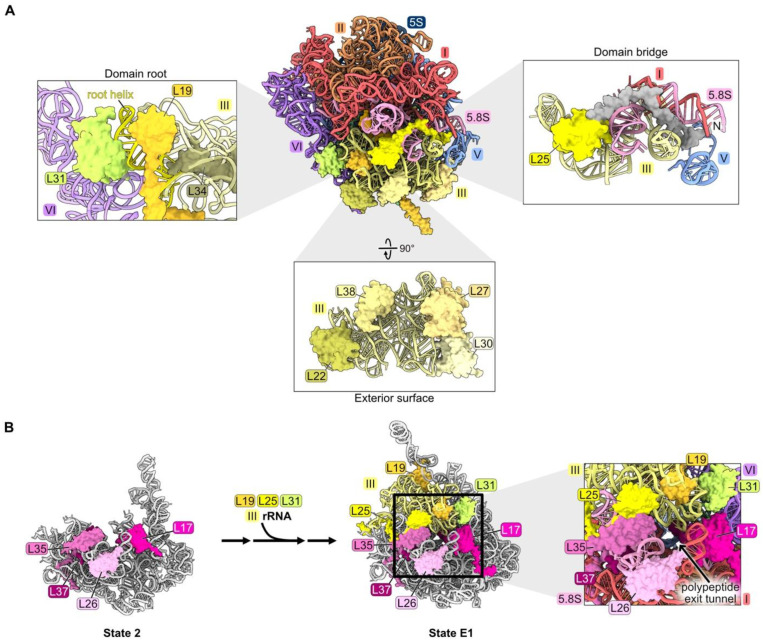
Domain III (RPs L19, L22, L25, L27, L30, L31, L34, and L38): (**A**) The locations of domain III RPs in mature LSU based on structural features. L19, L31, and L34 are shown embedded on the domain III root helix. L22, L27, L30, L38 are bound to the exterior surface of the domain III rRNA. The L25 globular domain (yellow) binds domain III rRNA, and the N-terminal extension of L25 containing amino acids 1–61 (grey) contacts neighboring rRNA domains (PDB: 4V88); (**B**) the completion of exit platform construction by assembly of RPs L19, L25, and L31 (PDB: 6C0F, 7NAC) [20,21].

**Figure 7 biomolecules-14-00975-f007:**
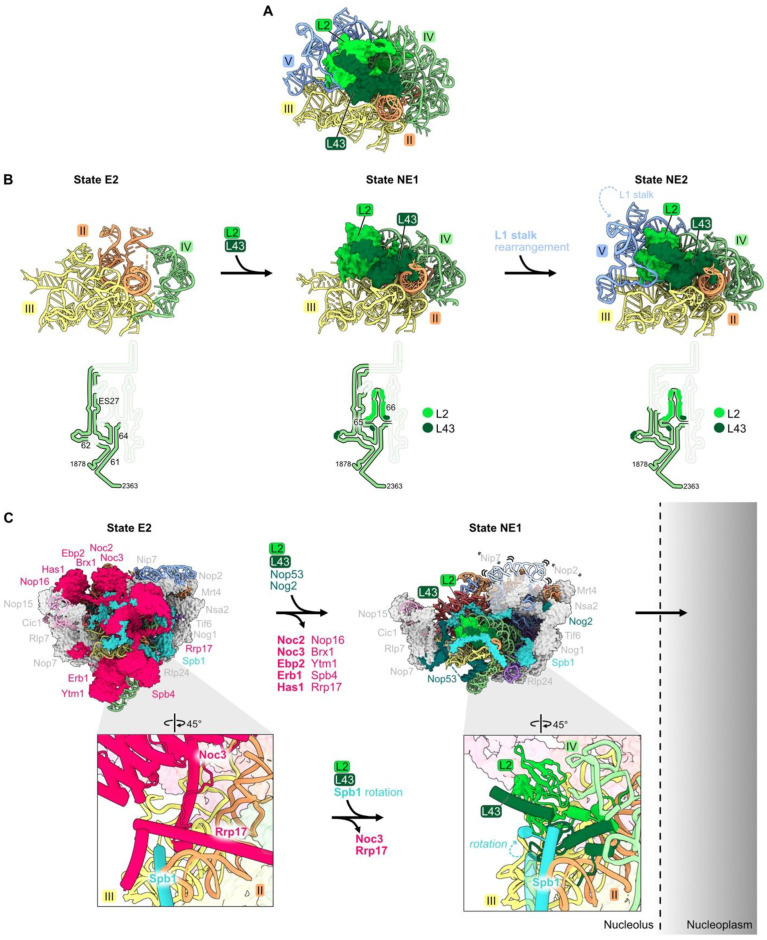
Domain IV (RPs L2 and L43): (**A**) L2 and L43 are positioned between rRNA domains II, III, IV, and V in the mature LSU (PDB: 4V88); (**B**) the entry of L2 and L43 enables compaction of rRNA domain IV and stabilizes the rotation of the L1 stalk (PDB: 7NAC, 7U0H, 6YLY) [21,28]. The bottom panel depicts changes in the stabilization of domain IV rRNA during this interval; (**C**) significant particle remodeling occurs during the entry of L2 and L43 just prior to the exit of pre-60S particles from the nucleolus to the nucleoplasm. AFs that contain intrinsically disordered domains and that exit at this interval are shown in bold. Note that Nip7, Nop2, and the L1 stalk are invisible in State NE1. (PDB: 7NAC, 7U0H).

**Figure 8 biomolecules-14-00975-f008:**
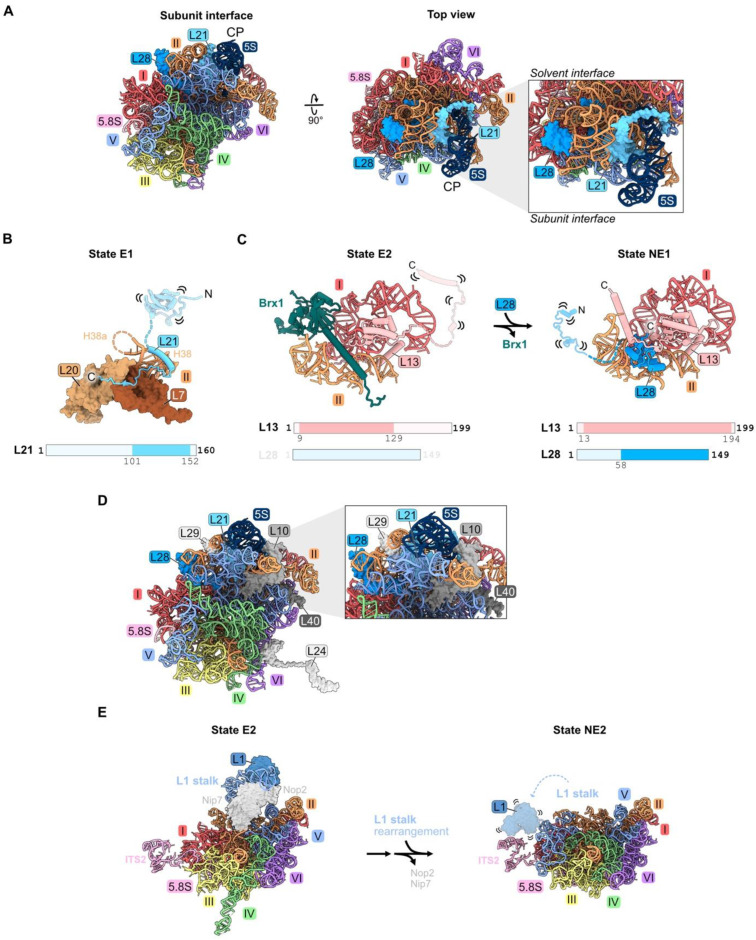
Domain V (RPs L1, L21, and L28): (**A**) The location of L21 and L28 in domain V beneath the central protuberance (CP) in the mature LSU. Note that L1 is not resolved in this structure (PDB: 4V88); (**B**) the C-terminal domain of L21 is stabilized on domain II RPs in State E1 (PDB: 7R7A); (**C**) a molecular switch: the exit of Brx1 enables the entry of the C-terminal globular domain of L28 (7NAC, 7U0H); (**D**) the assembly of L10, L24, L29, and L40 during nucleoplasmic and cytoplasmic stages of LSU construction is most affected by the absence of L21 or L28 (PDB: 4V88); (**E**) the rotation of the L1 stalk during the transition of the state E1 intermediate to state NE2. Note that L1 and H77-78 are not resolved following L1 stalk rotation in the nucleus. L1 and H77-78 are overlayed onto the NE2 structure (PDB: 7NAC, 6YLY).

**Figure 9 biomolecules-14-00975-f009:**
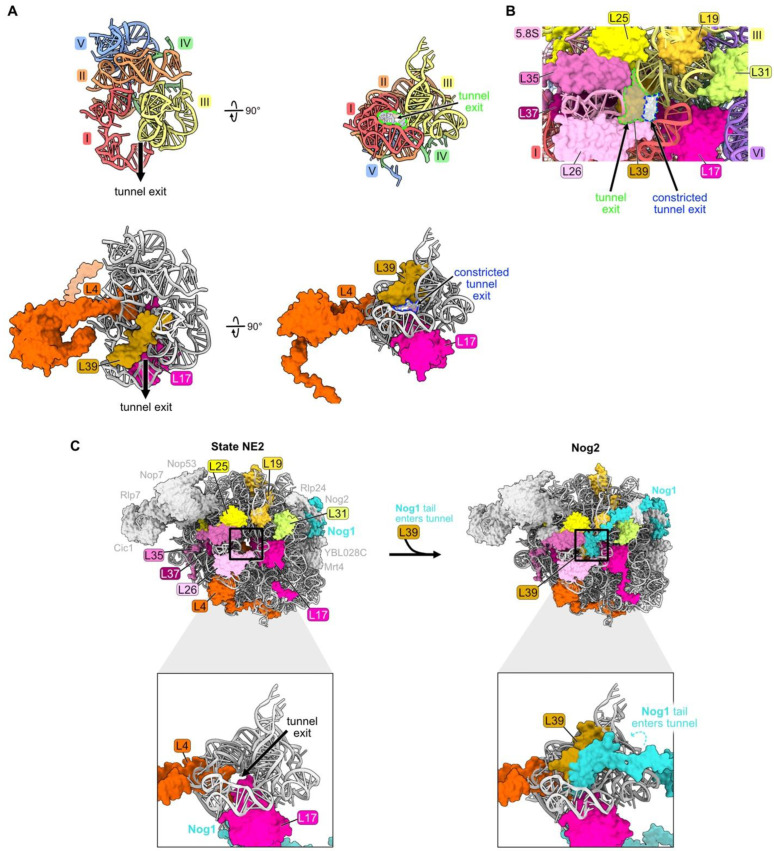
The polypeptide exit tunnel (RP L39): (**A**) The polypeptide exit tunnel is comprised of portions of rRNA domains I, II, III, IV, and V. Segments of L4 and L17 are inserted into the tunnel and L39 is embedded near the tunnel exit (PDB: 4V88); (**B**) the presence of L39 reduces the diameter of the tunnel exit. The space outlined in green highlights the size of the tunnel exit in the absence of L39. In contrast, the space outlined in blue indicates the reduced dimensions of the tunnel exit when L39 is present (PDB: 4V88); (**C**) particles before and after L39 entry, including RPs surrounding the tunnel exit. The C-terminal tail of Nog1 enters the tunnel coincident with L39 in the Nog2 assembly intermediate (PDB: 6YLY, 3JCT) [28,70].

**Figure 10 biomolecules-14-00975-f010:**
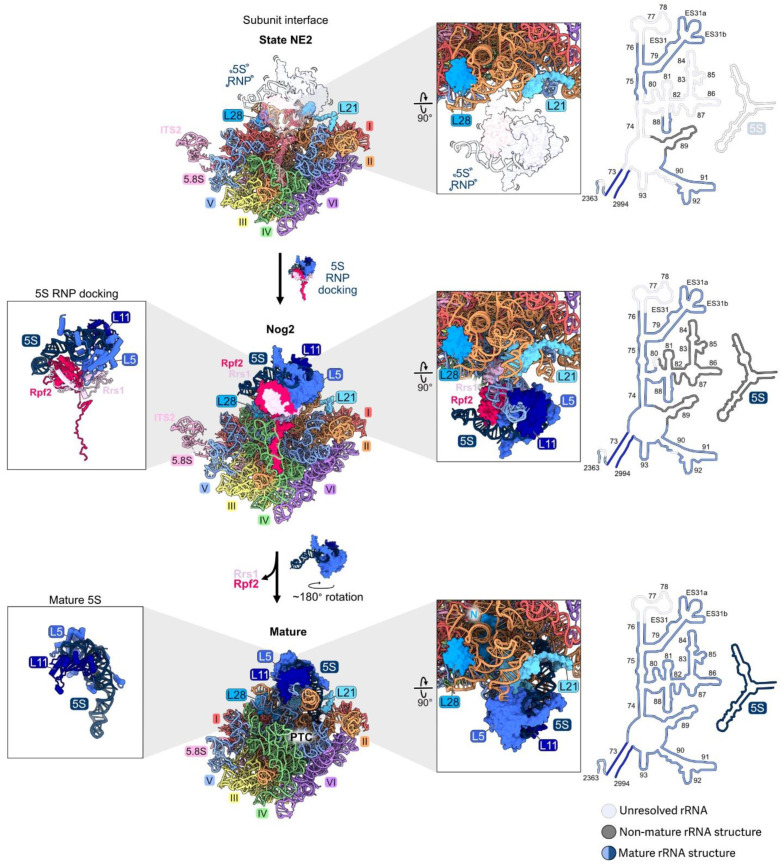
The central protuberance (RPs L5, L11): RPs L5 and L11 bind to the separately transcribed 5S rRNA and dock onto domain V on top of the LSU to form the central protuberance. The 5S RNP assembles with pre-ribosomes early, but is not visible by cryo-EM until the transition from state NE2 (top) to the Nog2 particle (middle). The 5S RNP undergoes ~180° rotation to form the mature structure (bottom) (PDB: 6YLY, 3JCT, 4V88) [28,47,70]. Abbreviation: peptidyltransferase center, PTC. Adapted from [28].

## Data Availability

Not applicable.

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
