# Peer review of "Putting It All Together: The Roles of Ribosomal Proteins in Nucleolar Stages of 60S Ribosomal Assembly in the Yeast *Saccharomyces cerevisiae"

_biomolecules, 2024, doi:10.3390/biom14080975_

Round 1

Reviewer 1 Report

Comments and Suggestions for Authors

This is a timely, high quality review from a leader in the field on the roles of ribosomal proteins in nucleolar stages of 60S ribosomal assembly in Saccharomyces cerevisiae, interpreted in the context of recent cryo-electron microscopy structural analysis. The manuscript is clearly written, comprehensive and accurate. The authors indicate several avenues for future research. It contains a large amount of detailed information but reads well and benefits from multiple informative figure panels. I am sure it will be widely read.

Minor point

Why in Figure S2C the authors have included L41 as a cytoplasmic large subunit assembly factor given that structural studies have shown that in fact L41 is a small subunit protein (eS32) that forms a bridge at the interface of the two subunits?  I did not see any other reference to L41 in the main text.

Author Response

Comment 1: This is a timely, high quality review from a leader in the field on the roles of ribosomal proteins in nucleolar stages of 60S ribosomal assembly in Saccharomyces cerevisiae, interpreted in the context of recent cryo-electron microscopy structural analysis. The manuscript is clearly written, comprehensive and accurate. The authors indicate several avenues for future research. It contains a large amount of detailed information but reads well and benefits from multiple informative figure panels. I am sure it will be widely read.

Response 1: We thank reviewer 1 for the positive feedback.

Comment 2: Why in Figure S2C the authors have included L41 as a cytoplasmic large subunit assembly factor given that structural studies have shown that in fact L41 is a small subunit protein (eS32) that forms a bridge at the interface of the two subunits?  I did not see any other reference to L41 in the main text.

Response 2: Thank you for pointing this out to us. We have now included in the legend of Figure S2C the following sentence. "Note that L41 is visualized in a few fungal LSUs, but otherwise is only observed in the SSU of archaea and eukaryotes, and thus is designated as eS32 in these organisms (N. Ban, unpublished)."

Reviewer 2 Report

Comments and Suggestions for Authors

Classical ribosomology has long ago come to the conclusion that during the biogenesis of ribosomes, both in vivo and in vitro ribosomal proteins  carry out a specific process of folding ribosomal RNA macromolecules into specific unique ribonucleoprotein structures. Thus  directly or indirectly, they participate in the formation of the functional centres of the ribosome. The secondary structure of rRNA, usually determined by combination of phylogenetic and biochemical methods, is here  largely preserved.

In recent years, rapid progress in single-molecule technology, and especially cryo-EM, has made it possible to study the structure, function and biogenesis of complex biological macromolecular complexes at a near-atomic resolution. Perhaps this primarily applies to ribosomes. This fact is demonstrated in a clear, concise and very convincing form in the authors' peer-reviewed manuscript entitled “ Putting It All Together: The Roles of Ribosomal Proteins in Nucleolar Stages of 60S Ribosomal Assembly in the Yeast Saccharomyces cerevisiae”.

Although the corresponding author of this manuscript has published a large number of experimental papers and at least four reviews in recent years on the biogenesis of yeast ribosomes, this problem is addressed in the manuscript from a new angle. Its novelty is beyond doubt: it perfectly reflects and fully corresponds to the new level of study of ribosomes. It is of interest not only to those who are involved in the biogenesis of this given object, but to everyone working in the field of studying the mechanism of protein synthesis in eukaryotic cells. Moreover, it is equally important both for professional ribosomologists and for researchers just starting to work in this field.

It is important that the text of the manuscript is illustrated with drawings of exceptionally high quality.

Comments for the  Authors.

In section 6, Going Forward, I would add one more problem. “What is the mechanism for transmitting  conformational signals about structural changes caused by a protein in a given region of rRNA to its neighbouring and distant regions?”

Minor comment. On page 6 please clarify that “pre-7S rRNA” is the precursor of 5.8 S rRNA.

Author Response

Comment 1: In section 6, Going Forward, I would add one more problem. “What is the mechanism for transmitting  conformational signals about structural changes caused by a protein in a given region of rRNA to its neighbouring and distant regions?”

Response 1: We thank reviewer 2 for their helpful suggestions. We agree that this is a critical question that remains largely unanswered. We have added the following to the list of questions in the Going Forward section. "How are structural changes transmitted to proximal and distal rRNP neighborhoods in the pre-ribosome as assembly progresses?"

Comment 2: Minor comment. On page 6 please clarify that “pre-7S rRNA” is the precursor of 5.8 S rRNA.

Response 2: Thank you, we made this improvement as follows: "...processing of the 7S pre-rRNA, a precursor of 5.8S rRNA, ..."

Reviewer 3 Report

Comments and Suggestions for Authors

This review article by Ayers and Woolford (biomolecules-3136757) summarizes the current knowledge about the assembly of ribosomal proteins into pre-ribosomal particles during the nucleolar phase of 60S subunit assembly. They describe in detail how the stepwise association or ribosomal proteins promotes sequential rRNA folding and helps to mould consecutive pre-60S particles along the maturation path.

This is a highly accomplished review article that I enjoyed very much reading. Notably, the authors manage to present this highly complex process in a simple and understandable manner. The well-written text is supported by high-quality figures that nicely illustrate the structural changes brought about by the binding of ribosomal proteins to the different pre-60S intermediates for which cryo-EM structures are available. As a take-home message, the authors also highlight the key mechanisms by which the association of ribosomal proteins drives ribosome assembly forward.

I have only a few minor points, as well as some suggestions for corrections, that the authors might consider before publication:

1)      Abstract, second sentence: To make the sentence clearer, remove the comma before “as well as” and insert “on” before “the assembly”.

2)      page 2: binds instead of bind in the sentence “As rRNA is transcribed, it begins to fold into its secondary structure and binds to RPs and AFs.”

3)      page 11: The statement “Depletion of L3 results in accumulation of 27SA2 pre-rRNA, …” is, in my opinion, not supported by the published data. The primer extension analysis reported in reference 32 shows similar 27SA2 levels in wild-type and L3-depleted cells, while the steady-state levels of the 27SB pre-rRNAs are strongly reduced. The analysis of pre-rRNA processing kinetics by pulse-chase labelling, reported in reference 46, shows that less 27SA2 is formed and that generation of stable 27SB pre-rRNAs cannot be observed upon L3 depletion. Similarly, an earlier report (Rosado et al. Nucleic Acids Research 2007) also showed that L3 depletion has only a minor effect on 27SA2 levels, while 27SB levels are strongly reduced.

4)      page 19: The appropriate and original reference for the statement “First, the dedicated chaperone Syo1 binds L5 and L11 in the cytoplasm and facilitates their import into the nucleus.” is Kressler et al. Science 2012.

5)      page 21; section title: assemble instead of assembly

6)      reference 24: the publication year is 2024 and not as indicated 2023

7)      references 70 and 71: incomplete author lists

8)      The resolution of most main figures is rather low in the PDF file that was provided for reviewing. This issue should be fixed as it would be a pity to display these great figures in a poor quality.

9)      Figure S3: Mention in the legend that the indicted association time point of RPs and AFs is based on their visualization in cryo-EM structures of the different pre-60S intermediates and that their actual or initial entry point may differ, for example Ssf1 associates earlier than Nsa1, but can only be visualized in State 2 pre-60S particles. Maybe, it would also help to clearly state early on in the text that there can be differences between the composition of purified pre-ribosomal particles and what can actually be visualized in the cryo-EM structures, presumably due to structural heterogeneity and/or flexibility.

10)   Figure S3: Only the entry point but not the exit point of Nop2 is indicated.

Author Response

Comment 1: Abstract, second sentence: To make the sentence clearer, remove the comma before “as well as” and insert “on” before “the assembly”.

Response 1: Thank you for catching this grammatical error. We have revised the sentence to the following: "We summarize the effects of depleting RPs on pre-rRNA processing and turnover, on the assembly of other RPs, and on the entry and exit of assembly factors (AFs)."

Comment 2: page 2: binds instead of bind in the sentence “As rRNA is transcribed, it begins to fold into its secondary structure and binds to RPs and AFs.”

Response 2: Thank you. We have changed the sentence to the following: "As rRNA is transcribed, it begins to fold into its secondary structure and binds to RPs and AFs (Figure S2A)."

Comment 3: page 11: The statement “Depletion of L3 results in accumulation of 27SA2 pre-rRNA, …” is, in my opinion, not supported by the published data. The primer extension analysis reported in reference 32 shows similar 27SA2 levels in wild-type and L3-depleted cells, while the steady-state levels of the 27SB pre-rRNAs are strongly reduced. The analysis of pre-rRNA processing kinetics by pulse-chase labelling, reported in reference 46, shows that less 27SA2 is formed and that generation of stable 27SB pre-rRNAs cannot be observed upon L3 depletion. Similarly, an earlier report (Rosado et al. Nucleic Acids Research 2007) also showed that L3 depletion has only a minor effect on 27SA2 levels, while 27SB levels are strongly reduced.

Response 3: Thank you, we apologize for failing to properly describe the results from these three papers. We have now revised this paragraph as follows and added the reference to Rosado et al.: "Pulse-chase and steady-state assays of L3-depleted cells, reveal only a minor effect on amounts of 27SA2 pre-rRNA compared to wild-type cells. In contrast, levels of 27SB pre-rRNA are strongly reduced."

Comment 4: page 19: The appropriate and original reference for the statement “First, the dedicated chaperone Syo1 binds L5 and L11 in the cytoplasm and facilitates their import into the nucleus.” is Kressler et al. Science 2012.

Response 4: We apologize for using the wrong reference. We have now properly cited this sentence with Kressler et al.

Comment 5: page 21; section title: assemble instead of assembly

Response 5: We have changed the section title as indicated.

Comment 6:  reference 24: the publication year is 2024 and not as indicated 2023

Response 6: We have corrected this mistake. Thank you.

Comment 7:  references 70 and 71: incomplete author lists

Response 7: Thank you, we have corrected these references.

Comment 8: The resolution of most main figures is rather low in the PDF file that was provided for reviewing. This issue should be fixed as it would be a pity to display these great figures in a poor quality.

Response 8: We agree, and have asked the journal to correct this.

Comment 9: Figure S3: Mention in the legend that the indicted association time point of RPs and AFs is based on their visualization in cryo-EM structures of the different pre-60S intermediates and that their actual or initial entry point may differ, for example Ssf1 associates earlier than Nsa1, but can only be visualized in State 2 pre-60S particles. Maybe, it would also help to clearly state early on in the text that there can be differences between the composition of purified pre-ribosomal particles and what can actually be visualized in the cryo-EM structures, presumably due to structural heterogeneity and/or flexibility.

Response 9: We completely agree with this very important point and forgot to include it. We have now inserted the following sentence in the legend of Figure S3: "Note that some proteins may be present at other points in assembly but may not be resolved in a structural intermediate." We also clarified this point in the text in section 3.1 Investigating when each RP assembles into pre-ribosomes as follows: "In this review, we indicate the point of entry for each RP based upon when it is first resolved in pre-ribosome intermediates via cryo-EM. Note that biochemical assays and cryo-EM structures of pre-ribosomes do not always agree. Failure to visualize a protein in a particle may result from structural heterogeneity and/ or flexibility of that protein."

Comment 10: Figure S3: Only the entry point but not the exit point of Nop2 is indicated.

Response 10: Thank you for finding this omission. We corrected this along with others that we noticed in this figure.